# Effect of Temperature on S32750 Duplex Steel Welded Joint Impact Toughness

**DOI:** 10.3390/ma16124432

**Published:** 2023-06-16

**Authors:** Srđa Perković, Aleksandar Sedmak, Zoran Radaković, Zijah Burzić, Simon Sedmak, Ljubica Radović, Jovana Mandić

**Affiliations:** 1Military Technical Institute, 11000 Belgrade, Serbia; perkovic.srdja@gmail.com (S.P.); zijah.burzic@vti.vs.rs (Z.B.); ljubica.radovic@mod.gov.rs (L.R.); jmandic96@gmail.com (J.M.); 2Faculty of Mechanical Engineering, University of Belgrade, 11120 Belgrade, Serbia; asedmak@mas.bg.ac.rs (A.S.); zradakovic@mas.bg.rs (Z.R.); 3Innovation Centre, Faculty of Mechanical Engineering, 11120 Belgrade, Serbia

**Keywords:** S32750 duplex steel, impact toughness, temperature, crack initiation energy, crack propagation energy

## Abstract

The search for alternative materials that can be used for parts of aircraft hydraulic systems has led to the idea of applying S32750 duplex steel for this purpose. This steel is mainly used in the oil and gas, chemical, and food industries. The reasons for this lie in this material’s exceptional welding, mechanical, and corrosion resistance properties. In order to verify this material’s suitability for aircraft engineering applications, it is necessary to investigate its behaviour at various temperatures since aircrafts operate at a wide range of temperatures. For this reason, the effect of temperatures in the range from +20 °C to −80 °C on impact toughness was investigated in the case of S32750 duplex steel and its welded joints. Testing was performed using an instrumented pendulum to obtain force–time and energy–time diagrams, which allowed for more detailed assessment of the effect of testing temperature on total impact energy and its components of crack initiation energy and crack propagation energy. Testing was performed on standard Charpy specimens extracted from base metal (BM), welded metal (WM), and the heat-affected zone (HAZ). The results of these tests indicated high values of both crack initiation and propagation energies at room temperature for all the zones (BM, WM, and HAZ) and sufficient levels of crack propagation and total impact energies above −50 °C. In addition, fractography was conducted through optical microscopy (OM) and scanning electron microscopy (SEM), indicating ductile vs. cleavage fracture surface areas, which corresponded well with the impact toughness values. The results of this research confirm that the use of S32750 duplex steel in the manufacturing of aircraft hydraulic systems has considerable potential, and future work should confirm this.

## 1. Introduction

Impact toughness is one of the most important material properties and was defined by Charpy more than 120 years ago as the total energy to break a notched specimen with a single pendulum strike [1,2]. Later on, instrumented pendulums capable of recording force vs. time enabled separation of the total energy into the energy for crack initiation and for crack propagation, as described in [3]. One should keep in mind that materials, especially welded joints, exhibit completely different behaviour in the case of dominant crack initiation energy from materials with dominant crack propagation energy, even though they have the same total energy [3].

Although not directly, the results of Charpy instrumented pendulum testing can be used in the scope of structural integrity assessment procedures, as explained in the case of estimating fracture toughness values, either through the master curve concept [4] or by micro-mechanical material modelling [5]. In addition, the application of Charpy V instrumented testing as a tool for structural integrity assessment is presented in [2,6].

As already mentioned, welded joints are of special interest when toughness is analysed because both the energies for crack initiation and propagation vary significantly in the case of heterogeneous material consisting of base metal (BM), welded metal (WM), and a heat-affected zone (HAZ), including its subzones (coarse grain (CG) and fine grain (FG)) [7,8,9]. The ductile–brittle transition is another important aspect of welded joints due to their heterogeneity, as shown by 3D transient analyses of Charpy impact specimens [10]. In the case of two high-temperature steels investigated in [8,11,12], detrimental effects of complex welded joint microstructures have been shown. Two steels have been tested—A 387 Gr. B and A387 Gr. 91—both indicating higher resistance to crack initiation and propagation in the HAZ than in WM [8,11,12].

Here, we consider S32750 duplex stainless steel (W. 1.4410) used for dynamically loaded components in the chemical and petrochemical industries, as well as in aeronautical engineering more recently. It is often used in aggressive corrosion environments with service temperature limits below 350 °C [13]. Duplex alloy is a descriptor of an alloy where two phases are present in significant quantities, as in the case of basic duplex stainless steel made of a ferritic/austenitic Fe-Cr-Ni alloy with 30–70% ferrite. During the 1980s, more highly alloyed duplex grades were developed to withstand more aggressive environments. The so-called superduplex grades contain about 25% Cr, 6–7% Ni, 3–4% Mo, 0.2–0.3% N, 0–2% Cu, and 0–2% W, as in the case of S32750 stainless steel designed for highly corrosive environments [13].

Recently, a special issue dedicated to advances in duplex stainless steels (DSSs) [14] presented mechanical properties and corrosion resistance in respect to composition and heat treatment, i.e., microstructures, [15,16], including welding innovations [17,18]. Special focus was placed on diffusive and diffusionless phase transformations due to the presence of the metastable austenitic phase and the instability of ferrite at high temperatures, potentially affecting corrosion resistance and mechanical properties.

Microstructures of DSS have also been analysed in respect to deformation processes, including electro-plastic effects, [19], hot deformation, [20], and numerical simulations of DIC-measured strains [21]. An investigation of the electro-plastic effect on four different DSSs was performed and presented in [19], characterised by optical microscopy, X-ray diffraction, and tensile properties (tensile strength, total elongation, uniform elongation, and yield stress). In an investigation presented in [20], the optimum range of deformation temperature (considering that both austenite and ferrite have different deformation behaviours due to their different morphological, physical, and mechanical properties) was determined. Local strain distribution in duplex stainless steel during tension, as obtained using the digital image correlation (DIC) technique, was presented in [21], with a finite element inversion of nanoindentation experiments on austenitic and ferrite phases in duplex stainless steel carried out to obtain the stress–strain response of the two phases. Further, based on the representative volume element (RVE) and the material parameters obtained from the finite element inversion method, the local stress and strain behavior of duplex stainless steel at a microscale was simulated numerically [21].

Generally speaking, S32750 DSS has good weldability and high mechanical strength, but its welded joints are still not investigated in detail, especially with respect to resistance to cracks. As an example of such an investigation, one can take the welding of dissimilar joints (S32750 DSS and 316L stainless steel) performed with a pulsed laser [18]. It was shown that the heat input (between 45, 90, and 120 J/mm) did not significantly affect the ferrite–austenite phase balance and the microhardness in the fusion zone [18].

The mechanical properties and microstructures of austenite-ferrite DSS welded joints made by hybridization (laser and GMAW) and SAW were investigated and presented in [22], indicating the presence of ferritic-austenitic microstructures both in the BM and WM, as well as high tensile strength (TS) and high ductility of both the WM and HAZ. A similar investigation was performed on laser-welded S32520 DSS, as presented in [23], indicating that process parameters affected tensile strength in a complex way due to the change in heat energy efficiency with changes in parameters. It was also shown that reducing the strength provided more ductile welded joints.

In this paper, welded joints made of S32750 DSS are tested at different temperatures on a Charpy instrumented pendulum to evaluate the resistance to crack initiation and propagation in BM, WM, and the HAZ. The results for the BM are already published in [24], indicating that S32750 DSS had high total impact toughness at all the tested temperatures. It was also shown that crack propagation energies were significantly higher than crack initiation energies at all the testing temperatures, which is a favourable distribution of energies from the point of view of structural integrity. Therefore, additional testing is performed on welded joints, as presented in this paper, to determine the distribution of energies in the WM and HAZ and to find out if it is also a favourable distribution. Furthermore, micrography and fractographic analyses through optical and scanning electron microscopy are performed on all the specimens to determine the ductile vs. cleavage fracture surface areas and to better explain the obtained values of energies for crack initiation and propagation.

## 2. Materials and Methods

### 2.1. Base Material

The material used in this work was a commercially produced duplex steel designated UNS S32750 (W. EN 1.4410). Its composition is given in Table 1, and its properties are in Table 2.

### 2.2. Welding

Welding was performed using a tungsten inert gas (TIG) procedure, which is typically used for the welding of high-alloyed and duplex steels. TIG 22/9/3 LN wire (designation of W 22 9 3 LN according to EN 12072 [25]) was used as a filler material due to its favourable mechanical properties and chemical composition, which was similar to the base metal. This filler material provided welded metal resistant to aggressive environments (especially against corrosion), which is of great importance for the integrity of welded joints in the chemical and oil industries, where duplex steels are mostly used. The chemical composition and mechanical properties of the filler material are given in Table 3 and Table 4, respectively.

For the welding of plates with dimensions of 200 × 150 × 25 mm, a K-groove was prepared that was 3.2 mm wide. Two different wire diameters were used: Ø2.4 mm for the root pass and Ø3.2 mm for the fill passes. Due to the geometry of the groove and in order to reduce residual stresses, plates were welded alternately from the upper and lower sides of the groove. Initially, welding parameters were determined based on recommendations and previous experience, and qualification of the suggested welding technology was performed on test plates. Test welded joints were tested using X-ray radiography, and defects were detected. The sample was discarded, welding parameters were changed, and qualification welding was performed again. In this case, radiography did not detect any defects, and this confirmed that the new welding technology was acceptable for further welding of this material. The final welding parameters are given in Table 5. The macrostructure of the weld joint is shown in Figure 1, indicating the WM, HAZ, and BM, as well as the root and fill passes.

The macrostructure of the welded joint is shown in Figure 1. All the characteristic zones are present: the base metal, HAZ, and welded metal. In the welded metal, all the passes can be identified. In the base metal, some heterogeneity can be observed in the centre of the cross-section. This heterogeneity originated during hot rolling. Plates welded in this work were 25 mm thick. After hot rolling, the cooling rate in the centre of the hot-rolled strip/plate was very low due to both a low heat transfer coefficient and large weight. It is well-explained that alloying elements redistribute, leading to heterogeneity and the formation of bands, as can be seen in Figure 1. This, in turn, makes the possibility of forming bands with enriched or depleted zones of alloying elements [26,27,28,29]. It has been proved that elevated contents of chromium and molybdenum can trigger the formation of second-phase particles located near the mid-thickness of a plate, mostly sigma-phase (FeCr) particles within the bands and a small volume fraction of chi phase (FeMoCr). These bands are partially re-soluted during welding in the vicinity of the welded metal, indicating that faster cooling does not allow the formation of a sigma phase [26,27,28,29].

The occurrence of banded structures, which can be observed in Figure 1, affected only the tensile strength, but only slightly since its orientation was along the band. However, this effect did not significantly influence toughness since the crack was perpendicular to the bonds. It also did not affect hardness, as can be seen from the results in Table 6. 

### 2.3. Microstructure and Fractography

The microstructures were characterised with a Leitz optical microscope. To reveal the grain structure, the samples were etched in mix of hydrochloric and nitric acid for 15 s. A scanning electron microscope (SEM, Jeol JSM 6610LV, Tokyo, Japan) was used to investigate the fracture surfaces of impact toughness specimens.

### 2.4. Impact Toughness Testing

Impact testing was performed according to the standard of SRPS EN ISO 148-1:2017 [30] using specimens according to SRPS EN ISO 9016:2013 [31], as shown in Figure 1.

Testing was performed on a SCHENCK TREBELL 150/300 J instrumented Charpy pendulum at room temperature (i.e., 20 °C) and at lower temperatures down to −80 °C for the BM, WM, and HAZ. Three specimens were extracted from each characteristic zone, with the crack tip positioned in the BM, WM, and HAZ. In the case of the HAZ, the specimen tip was located as close to the WM as possible (Figure 2), i.e., in the CGHAZ, as a representative of the lowest toughness in the HAZ. In the case of the WM specimen, the tip was located close to the centre line.

Welded plates were 25 mm thick, and after welding, they were cut in half along the thickness using a water jet to avoid additional heat input, which could lead to further microstructural transformations (cut line shown in Figure 3). Specimens for Charpy testing were extracted from the two halves of the plate, and notches were made from both sides (upper and lower).

Since the tests were performed using an instrumented Charpy pendulum, it was possible to separate the crack initiation and propagation energies and to evaluate the effect of the notch location on the impact properties. In this way, it was possible to determine the energies for initiating and propagating a crack, enabling a better understanding of the crack resistance of the tested material. Energy separation was performed according to the force maximum value so that the area to the left represented the energy for crack initiation, A_i_, and the area to the right the energy for crack propagation, A_p_ (Figure 4). Acquisition measuring methods were used according to standard SRPS EN ISO 14556:2020 [32].

## 3. Results

### 3.1. Microstructure

The microstructure of the base metal is shown in Figure 5a. It consisted of austenite (white) and ferrite (dark) phases. The austenite was present as laths inside the ferrite matrix. The microstructure in the centre of the weld showed austenite in the form of Widmannstätten plates, as depicted in Figure 5b. Austenite nucleated on the grain boundary of randomly oriented ferrite. The heat-affected zone showed slight microstructural changes (Figure 5c,d). The austenite grains were slightly coarser.

### 3.2. Charpy Testing

The results of the Charpy instrumented pendulum testing are shown in Table 7, Table 8 and Table 9 (BM, WM, and HAZ, respectively) and in the F–T diagrams (Figure 6, Figure 7 and Figure 8 for BM, WM, and HAZ, respectively), indicating the total energy and crack initiation and propagation energies at room and lower temperatures down to −80 °C. This temperature was selected as the transition temperature for the selected material based on previous experiments that were performed at temperatures up to −196 °C.

The results are presented as average values and deviations.

Based on these results, one can now determine the null ductility (transition) temperature (ND(T)T) corresponding to the total impact energy of 27 J, as shown in Figure 9, where total impact energy is plotted against temperature. As one can see from Figure 9, the ND(T)T was T_p_ = −110 °C for BM, while that for WM was −80 °C and HAZ was −50 °C.

### 3.3. Fractography

Macroscopic views of the surfaces of fractured specimens are given in Figure 10, Figure 11 and Figure 12 for the BM, WM, and HAZ, respectively, for all the tested temperatures. The results of the fractographic analysis are shown in the same way in Figure 13, Figure 14 and Figure 15.

Base metal: The fracture surfaces of impact test specimens in the temperature range from +20 °C to −80 °C are shown in Figure 10. Macroscopic observation showed extensive shear lips on both sides of the base metal specimens tested at +20 °C. Shear lips were also observed in all the specimens, even on the specimen tested at −80 °C, but they narrowed with decreasing temperature. Fracture areas covered by dimples were observed at all temperatures. Delamination in the centre of the specimen firstly occurred at −40 °C, while it was more pronounced at lower temperatures (−60 °C and −80 °C). It was noted that delamination was initiated on the notch tip (Figure 10c,d) and grew perpendicularly to the main crack. The same specimen’s notch tip area exhibited both ductile dimples (Figure 13a) and some triangular areas of mostly transgranular cleavage fracture (indicated by arrows in Figure 10c,d and Figure 13b,c at higher magnification). The latter feature was more pronounced at −80 °C (Figure 10d).

Welded metal: The fracture surfaces of impact test specimens at a temperature range from +20 °C to −80 °C are shown in Figure 11. Macroscopic observation showed shear lips at all temperatures. SEM observation showed that the fracture surface tested at +20 °C was covered by ductile dimples. Samples tested at −80 °C showed mixed-mode fractures, i.e., both dimples and cleavage (Figure 14).

Shear lips were also observed in specimens where notch tips were positioned in the HAZ, and the width decreased with decreasing temperature. Delamination was pronounced from −40 °C to −80 °C (Figure 12). A mixture of cleavage fracture and ductile dimple fracture was observed in the propagation zone at all temperatures (Figure 15). The fraction of cleavage fractures increased with decrease in the temperature.

## 4. Discussion

As one can see from Table 6, the total impact energy for the BM was well above the critical value, taken here as 27 J. Actually, the ND(T)T for the BM was −110 °C, as shown in Figure 9. In the case of the WM and HAZ, the ND(T)T amounts were significantly higher at −80 °C and −50 °C, respectively, indicating that welded joints made of S32750 duplex steel could operate safely at temperatures above −50 °C.

When considering crack initiation and propagation energies, one can see that, in all cases (BM, WM, and HAZ), the latter one dominated, providing relatively good resistance to crack propagation for the whole welded joint. Once again, the critical zone was the HAZ, with a crack propagation energy of only 21 J at −40 °C, while the WM had sufficient crack propagation energy above −60 °C. It is important to notice that dominant propagation energy is a better option for welded joints since one should not rely on crack initiation energy in the WM and HAZ because one should never take for granted that they are free of crack-like defects.

If the BM is compared with the WM and HAZ, not only were significant reductions in all the energy values obvious, especially in the HAZ, but also large deviations from the average values were evident in a few cases. Although this clearly indicated the effect of heterogeneous microstructures, it did not affect the minimum required toughness in any case.

From the BM fracture surfaces, one can see extensive shear lips on the specimens tested at +20 °C. Shear lips were also observed in other BM specimens, even those tested at −80 °C, but they narrowed with decreasing temperature. Fracture areas covered by dimples were observed at all the temperatures. Delamination in the centre of the specimens firstly occurred at −40 °C, while it was more pronounced at lower temperatures (−60 °C and −80 °C). Typically, delamination was initiated at the notch tip (Figure 10c,d) and grew perpendicularly to the main crack. As shown in Figure 13, the notch tip area of the BM specimens tested at −80 °C exhibited both ductile dimples (Figure 13a) and dominantly transgranular cleavage fractures at the delamination and triangular areas (Figure 13b–d). Delamination phenomena were also presented in [33].

Welded metal macroscopic observation showed shear lips at all temperatures that were less and less expressed as the temperature was reduced, as shown in Figure 11a–d. This observation was supported by SEM, indicating that the fracture surface tested at +20 °C was covered by ductile dimples, whereas samples tested at −40 °C, −60 °C, and −80 °C showed mixed-mode fractures, i.e., both dimples and cleavage (Figure 14).

Shear lips were also observed in specimens where notch tips were positioned in the HAZ, which also became smaller with decreasing temperature, as shown in Figure 12a–d. Delamination was pronounced from −40 °C to −80 °C (Figure 12b–d). A mixture of cleavage fractures and ductile dimple fractures was observed in the propagation zone at all temperatures (Figure 15). The fraction of cleavage fractures increased with decrease in temperature.

It is clear that fractographic features closely followed temperature effects on impact energies for all the welded joint zones since more and more cleavage fractures were present as temperatures became lower and lower.

## 5. Conclusions

Based on the presented results, one can conclude the following:The base material had a very high toughness at all the tested temperatures, from 297 J (+20 °C) to 82 J (−80 °C). Contrary to this, the total impact energy was significantly affected by WM and HAZ heterogeneity since it reduced from 229.3 (WM at +20 °C) to 26.5 J (WM at −80 °C) and from 87.7 (HAZ at +20 °C) to 17.6 J (HAZ at −80 °C).The transition temperature for BM was very low at −110 °C, but it was significantly higher for the WM and HAZ at −80 °C and −50 °C, respectively, indicating that welded joints made of S32750 duplex steel can operate safely only at temperatures above −50 °C.The distribution of energies was favourable since crack propagation was significantly higher than initiation energy.

The results of this research showed that welded joints made with S32750 duplex steel maintained satisfactory levels of toughness at temperatures as low as −50 °C. This suggests that the selected material is suitable for applications in the aircraft industry, specifically for hydraulic systems. Further research into this matter (including but not limited to the effects of fatigue, corrosion, etc.) should be conducted in order to completely verify this.

## Figures and Tables

**Figure 1 materials-16-04432-f001:**
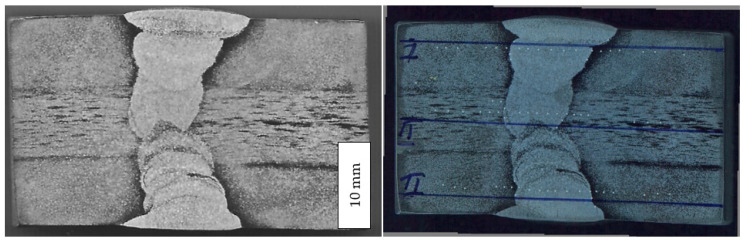
(**Left**) Macrostructure of welded joint. (**Right**) Hardness measuring locations.

**Figure 2 materials-16-04432-f002:**
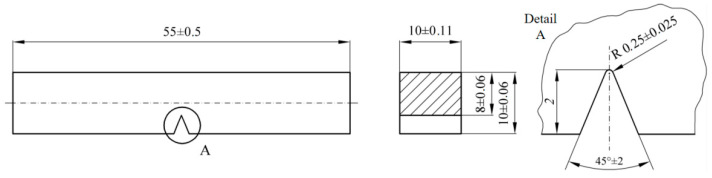
Shape and dimensions of a standard Charpy test specimen with a V-notch [31].

**Figure 3 materials-16-04432-f003:**
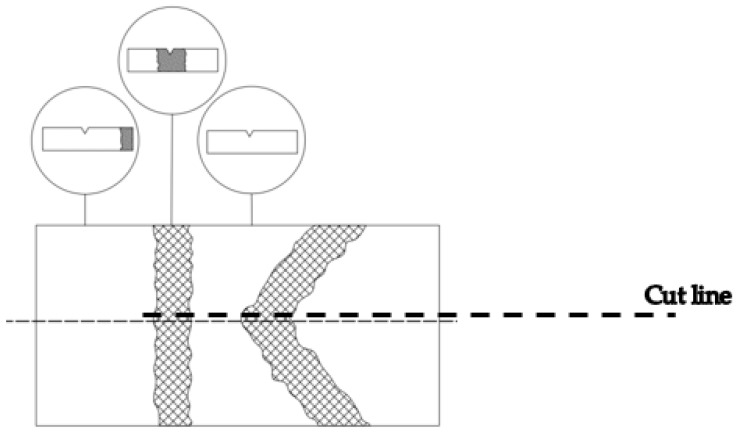
Welded joint Charpy specimen tip positioning with heat-affected zone thickness ranging from 3.5 to 4 mm in width.

**Figure 4 materials-16-04432-f004:**
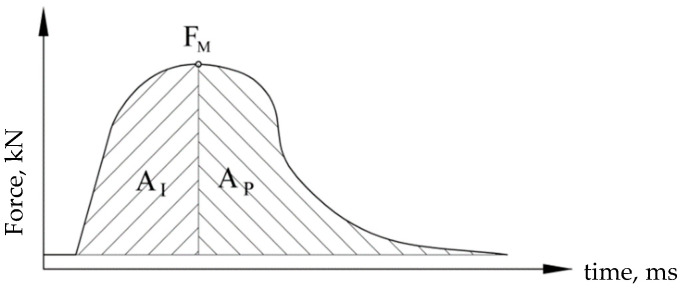
Scheme for separation of energies into crack initiation and propagation components.

**Figure 5 materials-16-04432-f005:**
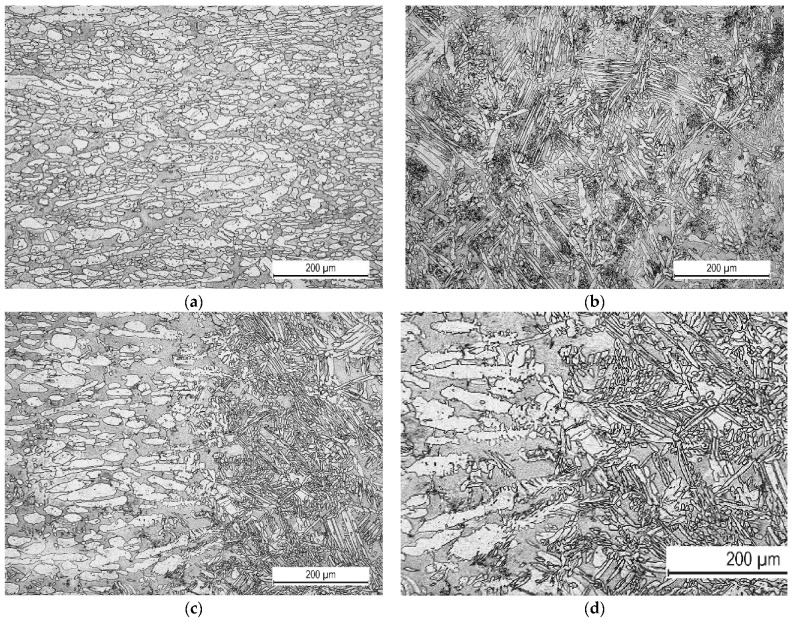
Microstructure of the welded joint: (**a**) base metal, (**b**) welded metal, (**c**,**d**) heat-affected zone.

**Figure 6 materials-16-04432-f006:**
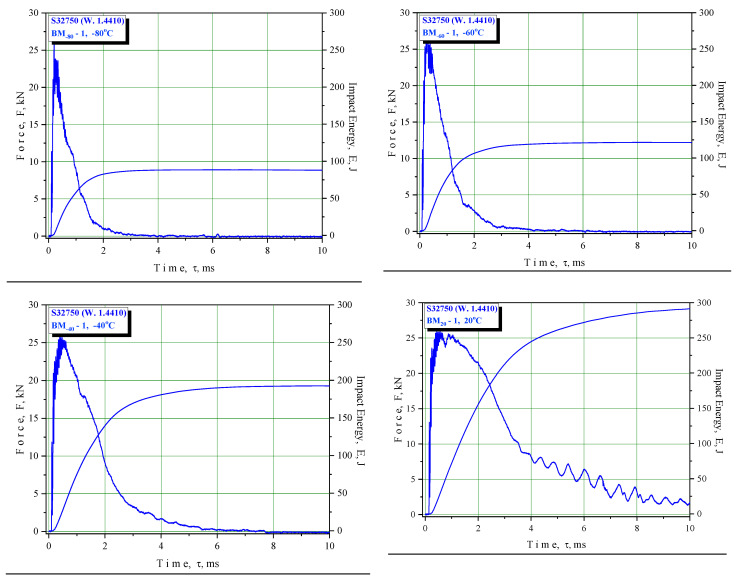
Relationships of force vs. time and energy vs. time for BM specimens.

**Figure 7 materials-16-04432-f007:**
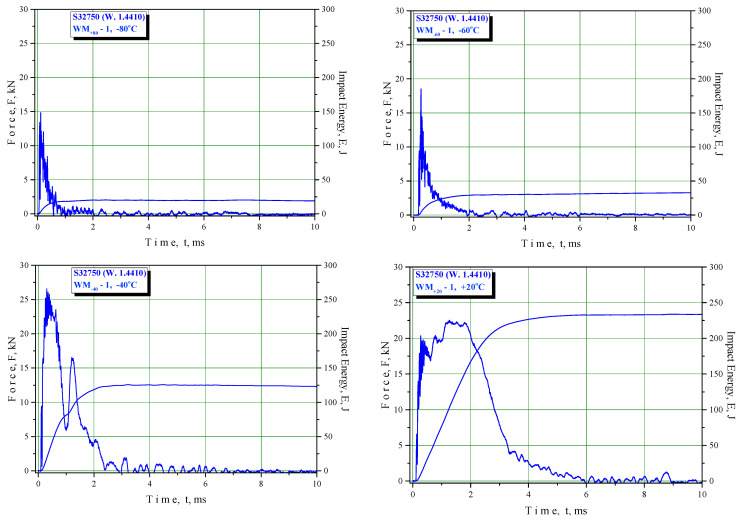
Relationships of force vs. time and energy vs. time for welded metal specimens.

**Figure 8 materials-16-04432-f008:**
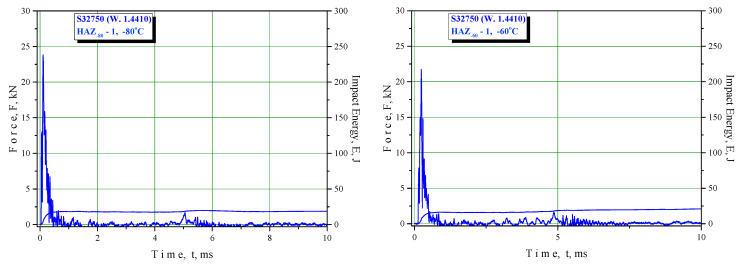
Relationships of force vs. time and energy vs. time for heat-affected zone specimens.

**Figure 9 materials-16-04432-f009:**
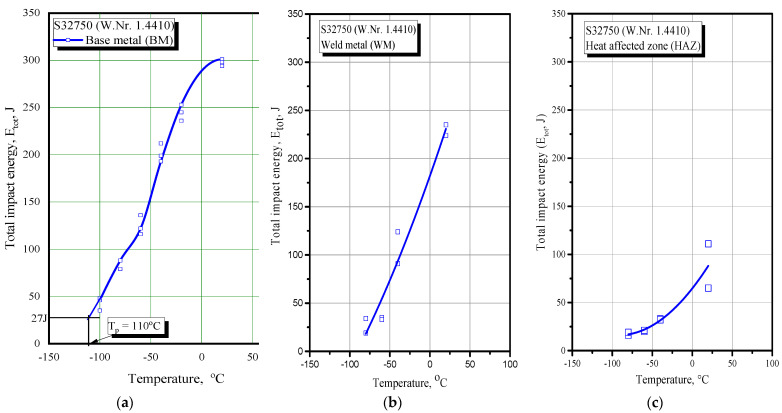
Total impact energy vs. temperature: (**a**) BM, (**b**) WM, (**c**) HAZ.

**Figure 10 materials-16-04432-f010:**
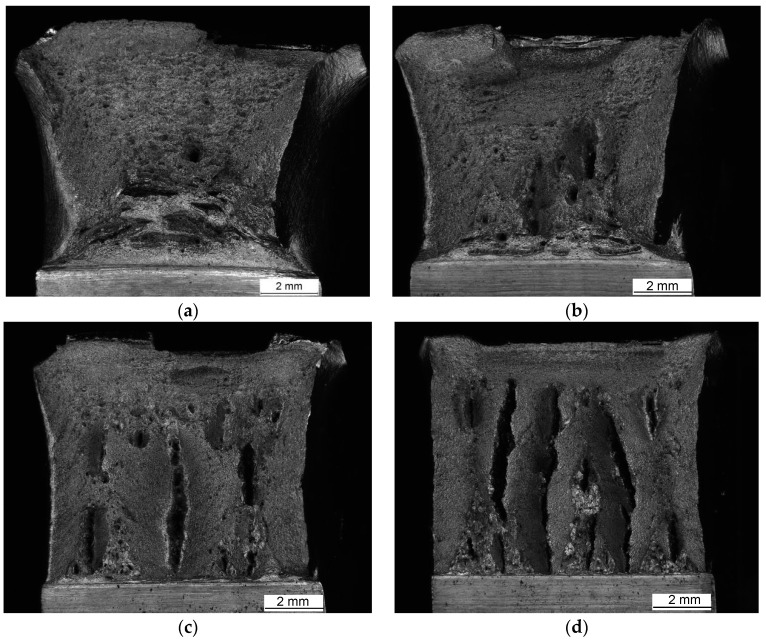
Fracture surfaces of BM tested at (**a**) +20 °C, (**b**) −40 °C, (**c**) −60 °C, and (**d**) −80 °C.

**Figure 11 materials-16-04432-f011:**
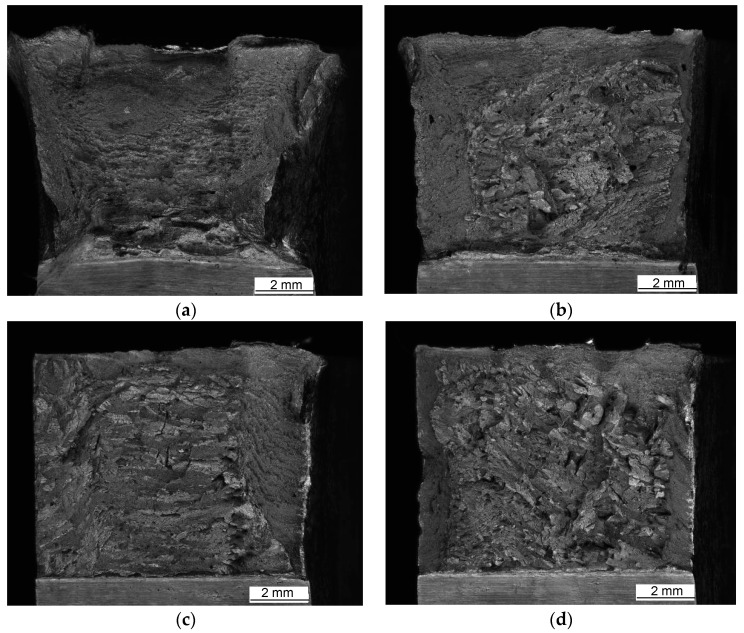
Fracture surfaces of WM tested at (**a**) +20 °C, (**b**) −40 °C, (**c**) −60 °C, and (**d**) −80 °C.

**Figure 12 materials-16-04432-f012:**
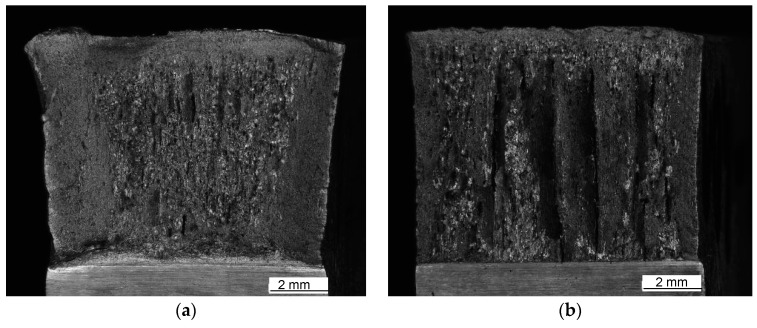
Fracture surfaces of the HAZ tested at (**a**) +20 °C, (**b**) −40 °C, (**c**) −60 °C, and (**d**) −80 °C.

**Figure 13 materials-16-04432-f013:**
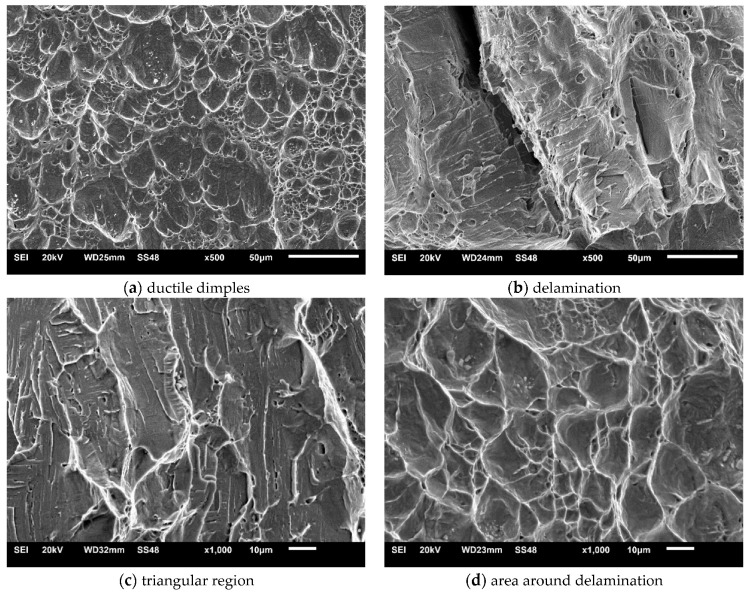
SEM microphotographs: fracture surfaces of BM tested at −80 °C.

**Figure 14 materials-16-04432-f014:**
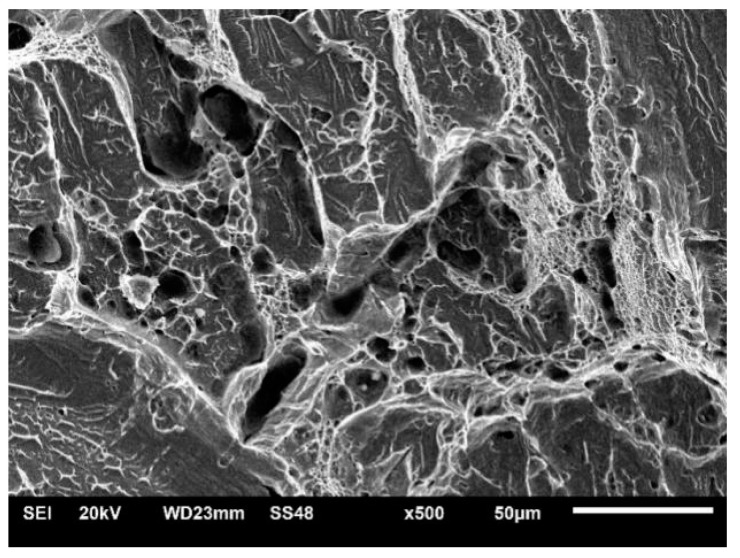
Mixed-mode fracturing of WM at −80 °C.

**Figure 15 materials-16-04432-f015:**
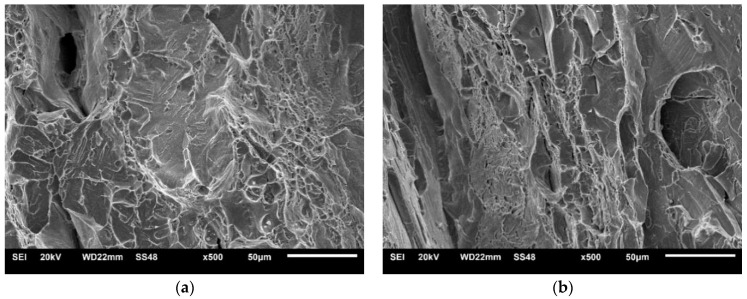
Mixture of cleavage fractures and ductile dimple fractures in the HAZ: (**a**) +20 °C and (**b**) −80 °C.

**Table 1 materials-16-04432-t001:** Chemical composition of the base material: superduplex stainless steel (S32750) (wt. %).

C	Si	Mn	S	P	Cr	Ni	Cu	Al	Mo	V	Ti	N	Co
0.02	0.49	0.88	0.0003	0.024	25.3	6.85	0.1	0.008	3.6	0.044	0.005	0.27	0.1

**Table 2 materials-16-04432-t002:** Mechanical properties of S32750 superduplex steel.

Yield Stress R_p0.2_, MPa	Tensile Strength R_m_, MPa	Elongation at Failure A_5_, %	Impact Toughness KV at 20 °C, J
584	855	32	297

**Table 3 materials-16-04432-t003:** Chemical composition of filler metal: TIG 22/9/3 LN (wt. %).

C	Si	Mn	Cr	Ni	Mo	N
<0.03	0.6	1.7	22.5	9.0	3.0	0.13

**Table 4 materials-16-04432-t004:** Mechanical properties of TIG 22/9/3 LN.

Yield Stress R_p0.2_, MPa	Tensile Strength R_m_, MPa	Elongation at Failure A_5_, %	Impact Toughness KV at 20 °C, J
>510	680–890	20	>47

**Table 5 materials-16-04432-t005:** Welding parameters for the first plate (3.2 mm K-groove).

Upper Side				Lower Side			
Pass	Current (A)	Voltage (V)	Duration (min)	Pass	Current (A)	Voltage (V)	Duration (min)
Root	170	14					
1	199	14	3:15	1	199	14	3:20
2	199	15	2:30	2	199	15	2:00
3	199	15.5	2:05	3	199	15.5	2:00
4	199	17	1:40	4	199	17	1:45
5	199	17.5	1:40	5	199	18	1:40

**Table 6 materials-16-04432-t006:** Hardness measurement results for all relevant welded joint regions.

Layer	Measurement No.		Hardness		HV 10			
		Zone (from Left to Right)						
		WM 1	HAZ 1	FL 1	WM	FL 2	HAZ 2	WM 2
I	1	437	435	437	433	435	433	435
	2	437	434	437	433	439	435	437
	3	435	433	437	431	437	433	437
	4				433			
	5				433			
	6				433			
II	1	437	435	437	435	441	437	435
	2	437	435	439	437	441	437	437
	3	435	435	437	437	439	437	437
	4				439			
	5				435			
III	1	437	435	437	435	437	435	435
	2	437	435	437	433	434	435	437
	3	437	435	437	433	435	437	437
	4				433			
	5				433			
	6				435			

**Table 7 materials-16-04432-t007:** Results for impact energies (average values and deviations) for BM.

Temperature, °C	Total Impact Energy E_tot_, J	Crack Initiation Energy, E_I_, J	Crack Propagation Energy, E_P_, J
−80	82 ± 6	17 ± 1	65 ± 5
−60	125 ± 11	25 ± 2	100 ± 9
−40	201 ± 11	45 ± 1	156 ± 10
+20	297 ± 3	52 ± 1	245 ± 3

**Table 8 materials-16-04432-t008:** Results for impact energies (average values and deviations) for WM.

Temperature, °C	Total Impact Energy, E_tot_, J	Crack Initiation Energy, E_I_, J	Crack Propagation Energy, E_P_, J
−80	26.5 ± 7.3	8.5 ± 4.4	18.0 ± 2.0
−60	34.1 ± 1	7.2 ± 2.9	26.9 ± 1.9
−40	112.4 ± 11.6	39.8 ± 10.7	67.6 ± 27.3
+20	229.3 ± 5.6	99.0 ± 3.3	130.3 ± 2.4

**Table 9 materials-16-04432-t009:** Results for impact energies (average value and deviation) for the HAZ.

Temperature, °C	Total Impact Energy, E_tot_, J	Crack Initiation Energy, E_I_, J	Crack Propagation Energy, E_P_, J
−80	17.6 ± 1.3	5.1 ± 0.9	12.5 ± 2.1
−60	20.3 ± 0.8	5.5 ± 0.7	14.8 ± 0.1
−40	32.3 ± 1.3	11.2 ± 1.6	21.1 ± 0.9
+20	87.7 ± 22.9	48.5± 22.0	39.2 ± 0.9

## Data Availability

Data is contained within the article.

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
