# Peer review of "Effect of Temperature on S32750 Duplex Steel Welded Joint Impact Toughness"

_materials, 2023, doi:10.3390/ma16124432_

Round 1

Reviewer 1 Report

1. Page 3, line 124, the authors incorrectly provided the chemical composition of the TIG 22/9/LN welding material.

2. Page 4, figure 1:

a) There is no description of the structure presented in Figure 1. The image shown in the figure exhibits deep structural heterogeneity. Please supplement the included photo with a description of the microstructure on the cross-section of the plate (what causes such significant heterogeneity?).

b) The figure clearly shows strong delimitation between the weld beads. Have the obtained welds undergone any NDT (Ultrasonic Testing or X-ray) examinations?

3. Page 5, Figure 3:

a) The authors stated that a K-groove was made, but Figure 3 shows an X-groove. Please amend the diagram presented in the figure.

b) There is no information in Figure 3 regarding the distance from the surface of the welded plate where the sample for testing was taken. Please provide this information.

4. In the article, the authors stated that the worst material properties are observed in the HAZ (Heat-Affected Zone). However, the authors only made this statement without providing a detailed analysis regarding, for example, the thickness of the zone, hardness, or unfavorable structural changes that affect it. Please provide additional information. As it is known, the HAZ does not have a constant width across the weld cross-section, so please include the values of the HAZ width on the weld cross-section.

5. The scale presented in Figures 10, 11, and 12 does not correspond to the actual dimensions provided on page 4, Figure 2. Please correct this.

6. The authors did not include microstructure after the impact process in the article. Therefore, please supplement the article with microstructures after the fracture process, observed in longitudinal section, especially for the HAZ, along with an analysis of the results, to demonstrate the initiation and propagation of cracks depending on the structural components.

7. After making the revisions mentioned in points 1-6, please supplement the conclusions presented in the article to reflect the introduced amendments. Currently, the article lacks scientific depth. The authors only stated that the material used in the research exhibits high resistance to low-temperature cracking, which is commonly known for the materials used in the study.

Reviewer 2 Report

The abstract should be written better and needs major revisions. The purpose of research and innovation should be clearly stated. Also, the performed tests should be presented first, and then the results should be presented quantitatively and qualitatively. The article needs general writing and grammar editing. The number of keywords can be increased. The introduction is written very briefly, and at the end, a suitable summary of the importance of the present issue is not provided.

The introduction needs to be reformed and deepened. Use the following resources to complete this section. Effect of heat input on interfacial characterization of the butter joint of hot-rolling CP-Ti/Q235 bimetallic sheets by Laser + CMT. Investigation of welding crack in micro laser welded NiTiNb shape memory alloy and Ti6Al4V alloy dissimilar metals joints. Effects of post-weld heat treatment on the microstructure and mechanical properties of laser-welded NiTi/304SS joint with Ni filler.

Tables 1-4 are presented on what basis? What is the basis of the experiment design and selected welding parameters (Table 5)? Figure 1 does not have a scale bar and labels for additional explanations. In this figure, different areas should be specified (WM, HAZ and BM, as well as root and fill passes ). On what basis was the temperature of -80 degrees Celsius chosen? Is there a special application at this temperature? The loading rate is one of the main parameters affecting the plasticity and energy absorption, which is not mentioned. Does Figure 4 need a reference? Has the fracture toughness been calculated? It is suggested to add the fracture toughness using the source below (3D printing of PLA-TPU with different component ratios: Fracture toughness, mechanical properties, and morphology). How many Charpy test samples were prepared for each group? How are the reproducibility of mechanical properties results checked? How accurate was the displacement measurement?

The results section is a report, and a more in-depth analysis should be added. It is suggested to use the following sources. Hydrogen embrittlement behavior of SUS301L-MT stainless steel laser-arc hybrid welded joint localized zones. Investigation of welding crack in micro laser welded NiTiNb shape memory alloy and Ti6Al4V alloy dissimilar metals joints. Microstructure section 3.1 is written very superficially and briefly. Different thermal regions should be identified, and more in-depth analyzes should be provided. Why is the force not converted into tension in diagrams 6-8? This section (Charpy testing) is also completely similar to the previous section and requires fundamental revision. It is suggested to add the discussion section to the results and combine these two sections. The images presented in Figures 13-15 are used in raw form. Additional explanations are not provided in the text. Add scale bar and label for additional description. It is suggested to modify the conclusion section as well as the abstract.

No comment.

Round 2

Reviewer 1 Report

No comments